# Heat Shock Protein Inhibitor 17-Allyamino-17-Demethoxygeldanamycin, a Potent Inductor of Apoptosis in Human Glioma Tumor Cell Lines, Is a Weak Substrate for ABCB1 and ABCG2 Transporters

**DOI:** 10.3390/ph14020107

**Published:** 2021-01-29

**Authors:** Nikola Pastvova, Petr Dolezel, Petr Mlejnek

**Affiliations:** Department of Anatomy, Faculty of Medicine and Dentistry, Palacky University Olomouc, Hnevotinska 3, 77515 Olomouc, Czech Republic; NikolaSkoupa@seznam.cz (N.P.); p.dolezel@atlas.cz (P.D.)

**Keywords:** tanespimycin, human glioma tumor cell panel, apoptosis, blood brain barrier, multidrug resistance, ABC transporters

## Abstract

Glioblastoma multiforme (GBM) is the most common primary brain tumor in adults and has a poor prognosis. Complex genetic alterations and the protective effect of the blood–brain barrier (BBB) have so far hampered effective treatment. Here, we investigated the cytotoxic effects of heat shock protein 90 (HSP90) inhibitors, geldanamycin (GDN) and 17-allylamino-17-demethoxygeldanamycin (17-AAG, tanespimycin), in a panel of glioma tumor cell lines with various genetic alterations. We also assessed the ability of the main drug transporters, ABCB1 and ABCG2, to efflux GDN and 17-AAG. We found that GDN and 17-AAG induced extensive cell death with the morphological and biochemical hallmarks of apoptosis in all studied glioma cell lines at sub-micro-molar and nanomolar concentrations. Moderate efflux efficacy of GDN and 17-AAG mediated by ABCB1 was observed. There was an insignificant and low efflux efficacy of GDN and 17-AAG mediated by ABCG2. Conclusion: GDN and 17-AAG, in particular, exhibited strong proapoptotic effects in glioma tumor cell lines irrespective of genetic alterations. GDN and 17-AAG appeared to be weak substrates of ABCB1 and ABCG2. Therefore, the BBB would compromise their cytotoxic effects only partially. We hypothesize that GBM patients may benefit from 17-AAG either as a single agent or in combination with other drugs.

## 1. Introduction

Cancer cells differ from normal cells in a number of ways. However, identification of the differences is not easy since there are more than one hundred organ specific types and subtypes of tumors [1]. Despite complex differences among tumor genotypes, there are common alterations in cell physiology that determine malignant phenotypes. Such alterations include: (a) sustained proliferative signaling; (b) insensitivity to antiproliferative signals; (c) resistance to cell death signals; (d) limitless reproductive potential; (e) reprograming of energy metabolism; (f) sustained angiogenesis; (g) tissue invasion and metastases; (h) evasion of the immune system; (i) tumor promoting inflammation; (j) genome instability and mutation [1,2].

In addition to these common features, there is increased dependence of tumor cells on heat shock proteins (HSPs) compared to nontumor cells. HSPs constitute a large family of molecular chaperones divided into six groups (HSP27, HSP40, HSP60, HSP70, HSP90, and large HSPs) according to molecular weights and functioning in diverse physiological and pathological conditions. Importantly, HSPs play a crucial role in cancer initiation, progression, metastasis, and resistance to radio- and chemotherapy [3,4]. Thus, it is not surprising that HSPs have become a therapeutic target for the treatment of tumors [5,6]. HSP90 is probably the best studied ATP-dependent molecular chaperone. Its inhibitors show significant antitumor effects. The natural product geldanamycin (GDN) exhibits strong antitumor effects in vitro, and its synthetic derivatives such as 17-allylamino-17-demethoxygeldanamycin (17-AAG, tanespimycin) have promising effects in clinical trials [6].

Glioblastoma multiforme (GBM) is the most common primary brain tumor in adults and has a poor prognosis. Patients with GBM usually die within 3 months if left untreated. Despite continuous improvement in GBM therapy currently consisting of surgical resection, radiation, and temozolomide (TMZ) treatments, almost all patients experience disease recurrence and the median duration of survival is 12–14 months [7,8]. No standard treatment option exists after recurrence. GBM is a genetically heterogeneous tumor with complex genetic alterations affecting signaling pathways that regulate cell proliferation, DNA repair, the cell cycle, metabolism, and cell adhesion, among others [8,9]. 

Despite the genetic heterogeneity of GBMs, some mutations occur very frequently and may include: epidermal growth factor receptor (*EGFR*), platelet-derived growth factor receptor (*PDGFR*), hepatocyte growth factor receptor (*HGFR*/*MET*), *O*-6-methylguanine-DNA-methyltransferase (*MGMT*), isocitrate dehydrogenase 1 and 2 (*IDH1*/*2*), cyclin-dependent kinases (*CDK*s), the *TP53* tumor suppressor gene, phosphatase and tensin homolog (*PTEN*), retinoblastoma transcriptional corepressor 1 (*PRB1*), phosphoinositide 3-kinase (*PI3K*), telomerase reverse transcriptase (*TERT*), serine/threonine-protein kinase B-Raf (*BRAF*), histone mutations (e.g., H3K27M), and codeletion of chromosomes 1p and 19q. These mutations cause alterations in key signaling pathways that drive the pathogenesis of glioblastoma. Most of these serve as attractive molecular targets for improving the treatment of GBM [10,11,12,13].

The availability of low molecular weight inhibitors and monoclonal antibodies that effectively target deregulated pathways identified in GBM and also used in other tumors, enables the development of new targeted therapies and testing of their relevance to GBM in in vitro models. A combination of appropriately chosen targeted therapies with TMZ may contribute to greater efficacy of GBM therapy [7,8,14]. However, data from clinical practice dampen exaggerated optimism. For example, expression analysis and preclinical data suggest that mutated receptor tyrosine kinase inhibitors (RTKIs) and other oncogenes are promising targets for low molecular weight inhibitors (e.g., erlotinib, imatinib) in the treatment of GBM, but the results from clinical trials suggest otherwise [7,8]. The latter produced promising results with bevacizumab, a single antiangiogenic monoclonal antibody; however, no clear conclusions can be drawn yet [14]. Recently, tumor immunotherapy has focused on toll-like receptors (TLRs) and their downstream signaling pathways. In particular, TLR-4 appears to be a promising target for immunotherapy of a variety of tumors, including GBM [15].

Here, we studied the cytotoxic effects of GDN and 17-AAG in six glioma tumor cell lines with different genetic alterations. We also addressed the question regarding how efficiently these drugs are effluxed out of cells by the main drug transporters, ABCB1 and ABCG2. We observed that GDN and 17-AAG are very efficient proapoptotic drugs that kill all studied glioma tumor variants. Our results further indicated that GDN and 17-AAG are weak substrates of ABCB1 and ABCG2. 

## 2. Results

### 2.1. Cytotoxic Effects of HSP90 Inhibitors in Glioma Tumor Cells

To determine the cytotoxic efficacy of the HSP90 inhibitors GDN and 17-AAG, we used glioma tumor cells with various mutations. Specifically, in this study we used a glioma tumor cell panel together with T98G cells. A brief characterization of these tumor cells is given in Table 1.

We observed that GDN and particularly 17-AAG exerted very high cytotoxic effects in glioma tumor cells irrespective of their genetic alterations. While GDN induced cell death at sub-micro-molar concentrations, 17-AAG induced cell death at nanomolar concentrations (Figure 1 and Table 2). Interestingly, T98G and H4 cell lines exhibited somewhat lower sensitivities to both HSP90 inhibitors (Figure 1, Table 1 and Table 2).

Next, we compared the cytotoxic efficacy of HSP90 inhibitors with TMZ, a drug currently used in combination with surgical resection and radiotherapy to treat glioblastoma in clinics [8,12,14]. Importantly, TMZ itself exhibited only moderate to low cytotoxicity in glioma tumor cells despite the decreased expression of *MGTM* (Table 3). Although lomeguatrib (LG) increased the cytotoxicity of TMZ in some cell lines, the IC_50_ values were still in the range of high micromolar concentrations (Table 3).

We further studied the mode of cell death induced by HSP90 inhibitors in glioma tumor cell lines, as this topic has not yet been studied in detail. Our results indicated that cell death induced by GDN and 17-AAG exhibited morphological and biochemical apoptotic hallmarks, including chromatin condensation and apoptotic body formation (Figure 2a–c), genomic DNA fragmentation (Figure 2d), and caspase3/7 activation (Figure 2e). 

### 2.2. Interactions of Geldanamycin and Tanespimycin with the Main Drug Transporters

The major drug transporters, ABCB1 and ABCG2, are markedly expressed in the blood–brain barrier (BBB), where they limit the permeability to xenobiotics, thereby reducing the cytotoxic effects of chemotherapy. In addition, both drug transporters can directly mediate the MDR phenotype if they are overexpressed in cancer cells [16]. Therefore, we measured the intracellular levels of GDN and 17-AAG in cells overexpressing both drug transporters. 

To do this, we used leukemia K562 cells with high and moderate expressions of ABCB1, K562/DoxDR3 and K562/DoxDR2, and ABCG2, K562/ABCG2CL10 and K562/ABCG2CL4, respectively (Figure 3). For comparison, we also measured the ABCB1 expression in glioma H4 cells (Figure 3). We observed that intracellular levels of GDN were significantly decreased in K562/DoxDR3 and K562/DoxDR2 cells (Figure 4b). Similarly, intracellular levels of 17-AAG were significantly decreased in both cells with high and moderate ABCB1 expressions (Figure 4c). Importantly, the overall efflux efficacy of both drugs was low (Figure 4). Specifically, it was five to six times lower than that of the calcein-AM fluorescent probe for evaluating ABCB1 function (Figure 4). Neither K562/ABCG2CL4 nor K562/ABCG2CL10 cells had significantly decreased intracellular levels of GDN (Figure 5b). However, only K562/ABCG2CL10 cells with high expression of ABCG2 had significantly reduced intracellular level of 17-AAG (Figure 5c). Here, the ability to decrease intracellular level of 17-AAG was approximately five times lower than that of pheophorbide A a fluorescent probe for assessment of ABCG2 function (Figure 5).

## 3. Discussion

Clinical practice has shown that GBM treatment is extremely difficult and, unfortunately, not successful. This is probably mainly due to the fact that the tumor is genetically very heterogeneous [11,12]. Standard GBM therapy consisting of tumor resection, radiotherapy, and TMZ has increased the median duration of survival to 12–14 months, but the long-term survival time is still less than 5 years [7,8]. Targeted treatment of GBM variants has not produced the expected results, probably because this particular tumor does not depend on a single signaling pathway that is amenable to such therapy [7,17]. In addition, any successful drug in the case of the brain needs to penetrate the BBB. 

Therefore, we applied a different approach in this study. We searched for a drug whose molecular target is common to all tumors and which can cross the BBB or at least have weak interactions with the main drug transporters. Tumors, including GBM, require HSPs to maintain the malignant phenotype. Of HSPs, HSP90 was found to be the key to proliferation, survival, invasion and metastases, and angiogenesis of GBM [17,18]. Therefore, we chose HSP90 inhibitors, GDN, and its derivative, 17-AAG (with fewer side effects) [19], to study their cytotoxic effects and mode of cell death in glioma tumor cell lines with different genetic alterations. In addition, we also focused on GDN and 17-AAG interactions with the main drug transporters ABCB1 and ABCG2.

Several initial studies suggested that HSP90 inhibitors, including 17-AAG, are very promising drugs with the potential for clinical use in the treatment of GBM [20,21,22,23,24]. Importantly, a number of critical client proteins of HSP90 in GBM have been identified [20,21,22,24]. However, to the best of our knowledge, none of the published work has systematically addressed the mode of cell death in response to HSP90 inhibitors. Similarly, GDN and its derivatives have been reported as substrates of ABCB1 [25,26]. However, no direct relationship between the expression of drug transporter and the intracellular concentration of HSP90 inhibitors has been established. In addition, to the best of our knowledge, no interaction between ABCG2 and HSP90 inhibitors has been described.

Here, we demonstrate that GDN and 17-AAG in particular are highly effective cytotoxic agents that kill all studied glioma tumor cell lines with various genetic alterations at sub-micro-molar and nanomolar concentrations (Figure 1, Table 2). Our results are in good agreement with those of other researchers, although these authors did not use a panel of glioma tumor cell lines [20,21,22,24]. It is necessary to note that the sensitivity of H4 and T98G cells to HSP90 inhibitors was somewhat lower than other glioma tumor cell lines (Figure 1, Table 1 and Table 2). While the decreased sensitivity to GDN and 17-AAG can be attributed to increased expression of ABCB1 in H4 cells (Figure 3, Table 2), no adequate explanation exists for T98G cells. In this case, we can only speculate about the activation of heat shock transcription factor 1 (HSF1) by HSP90 inhibitors, which induces increased expression of a number of molecular chaperones [26,27]. Some of these (e.g., HSP27) may then mediate resistance to HSP90 inhibitors [26]. However, the cytotoxic effect of both HSP90 inhibitors and in particular 17-AAG is still very high in both cell lines (Figure 1 and Figure 2, Table 2).

Our results further reveal that glioma tumor cell lines subjected to GDN and 17-AAG treatment die via cell death with the morphological and biochemical features of apoptosis (Figure 2). To our knowledge, the mechanism of the cell death mode has only been studied in detail in the T98G cell line and only in response to GDN [28]. We have no adequate explanation for the finding that 17-AAG is a more effective inducer of apoptosis than GDN in the studied glioma tumor cell lines (Figure 2 and Table 2). 

Importantly, the cytotoxicity of GDN and 17-AAG cannot be compared with that of TMZ or TMZ in combination with LG, whose cytotoxicity is approximately a thousand times lower (Table 2 and Table 3). The reason for the observed low cytotoxicity of TMZ or TMZ in combination with LG in the glioma tumor cell line is unknown. We can only speculate that the difference lies, among other things, in the fact that TMZ is administered in vivo in combination with radiotherapy [7,12]. The low efficacy of TMZ against glioma tumor cell lines, however, has also been reported by other authors [23].

ATP-binding cassette (ABC) transporters use the hydrolysis of ATP to translocate endogenous and exogenous small molecules across plasma and organellar membranes in a variety of cellular processes. A subset of 48 ABC transporters is implicated in drug resistance in some contexts. Of these, ABCB1 and ABCG2 appear to be the most important in the mediation of the MDR phenotype in cancer cells when overexpressed [16,29]. Importantly, drug transporter expression levels affect a range of drug resistance factors, including degree of resistance [30,31], inhibitor potency [32,33], and collateral sensitivity [34,35]. Furthermore, the physiological expression of drug transporters in tissues is often higher than that in tumor cells [32,36,37]. The H4 cell line accurately reflects this. The observed resistance to GDN and 17-AAG is low because it is mediated by low expression level of ABCB1 (Table 2, Figure 1, Figure 2 and Figure 3). 

ABCB1 and ABCG2 may also be indirectly involved in drug resistance since they affect the bioavailability of oral intake of drugs or limit drug uptake at the BBB [38]. With the above aspects in mind, we used chronic myeloid leukemia cells, K562/DoxDR2, K562/DoxDR3, K562/ABCG2CL4, and K562/ABCG2CL10, with moderate to high expression levels of drug transporters (Figure 3) since one can expect relatively high expression levels of ABCB1 and ABCG2 in the BBB [38]. 

Our results clearly show that although both inhibitors are substrates of ABCB1, the ability of ABCB1 to decrease intracellular levels of GDN and 17-AAG is approximately 45–65% of the level found in sensitive K562 cells (Figure 4). In other words, the ability of ABCB1 to efflux these inhibitors is approximately five to six times lower than that of calcein-AM, a well-characterized substrate of ABCB1 (Figure 4). The situation is even better in the case of the ABCG2 (Figure 5). Indeed, ABCG2 failed to decrease intracellular levels of GDN significantly in either cell. Intracellular levels of 17-AAG were significantly reduced only in cells with high ABCG2 expression and achieved 70–85% of the level found in sensitive K562 cells (Figure 5). The ability of ABCG2 to efflux 17-AAG was approximately five times lower than that of pheophorbide A, a well-characterized substrate of ABCG2 (Figure 5). These results suggest that both inhibitors of HSP90, GDN, and 17-AAG are weak substrates of ABCB1 and ABCG2. The observation that GDN and its derivatives are substrates of ABCB1 was originally reported by Huang et al. [25] and later studied by other researchers [26]. To the best of our knowledge, the interaction between ABCG2 inhibitors and HSP90 has not been described to date. Importantly, this study provides for the first time a quantitative aspect of the interaction between HSP90 inhibitors and the major drug transporters, ABCB1 and ABCG2.

Given that GDN cannot be used in clinics due to its side effects [18], we will further focus only on 17-AAG which is in the process of clinical trials to treat other cancers either as a monotherapy or in combination with other anticancer drugs [19]. Since (i) serum concentrations of 17-AAG achieve micromolar levels [39], (ii) IC50 values for 17-AAG do not exceed 50nM concentration in studied cell lines (Table 2), and (iii) ABCB1- and ABCG2-mediated efflux efficacy of 17-AAG is relatively low (Figure 4 and Figure 5), we hypothesize that 17-AAG can cross the BBB at concentrations that may be cytotoxic to GBM. Importantly, our results provide a mechanistic explanation for the findings of Sauvageot et al. who found that 17-AAG was capable of inhibiting intracranial tumors in mice [22], and those of Waza et al. who demonstrated that 17-AAG ameliorated polyglutamine-mediated motor neuron degeneration [40]. On the other hand, it should be recalled that other authors have found a rapid distribution of 17-AAG to all tissues except the brain [41]. Nevertheless, detailed analysis of published data shows that 17-AAG concentrations can reach micromolar concentrations in brain tissue [41]. It remains, however, that the substrate specificity of ABC transporters in humans and mice BBBs may not be the same. 

Despite the attractiveness of these results, they should be taken with caution, given that this is an in vitro study and all the results will need to be verified in clinical practice. For example, 17-AAG may not be able to cross the human BBB, and then several potential ways to circumvent it will need to be considered, such as direct settling in the surgical cavity or temporary opening of the BBB. Nevertheless, we believe that inhibition of HSP90 chaperone function is a promising strategy for the successful treatment of GBM. A number of new HSP90 inhibitors have been developed in recent decades, both on the basis of benzoquinone as well as nonbenzoquinone derivatives. Many of these have fewer side effects, their interaction with drug transporters is reduced, and they are currently in clinical trials [42]. In addition, some newly developed HSP90 inhibitors can simultaneously inhibit all members of the HSP90 family of proteins which increases their therapeutic potential [43].

## 4. Material and Methods

### 4.1. Chemicals and Cell Treatment

Geldanamycin (GDN) was purchased from Sigma-Aldrich (St. Louis, MO, USA). 17-Demethoxy-17-(2-propenylamino)-geldanamycin (17-AAG, tanespimycin) was obtained from Cayman (1180 East Ellsworth RD Ann Arbor, Michigan 48108, USA). Both inhibitors were dissolved in DMSO. The final concentration of DMSO in culture medium was approximately 0.1%. Zosuquidar trihydrochloride (ZSQ; LY335979), a potent inhibitor of ABCB1 transporter, and lomeguatrib (LG; 6-[(4-bromo-2-thienyl)methoxy]-9*H*-purin-2-amine), a potent inhibitor of MGMT, were purchased from Selleckchem (Houston, TX, USA). Ko143 (3S,6S,12aS)-1,2,3,4,6,7,12,12a-octahydro-9-methoxy-6-(2-methylpropyl)-1,4-dioxopyrazino-[1′,2′:1,6] pyrido [3,4-b]indole-3-propanoic acid 1,1-dimethylethyl ester, a potent and selective inhibitor of the ABCG2 transporter, was obtained from Enzo Life Sciences AG (Lausen, Switzerland).

### 4.2. Cell Culture

In this study, we used a human glioma tumor cell panel (TCP-1018) containing A172 (CRL-1620), SW1088 (HTB-12), H4 (HTB-148), U118-MG (HTB-15), and U87-MG (HTB-14) cell lines with varying degrees of genetic complexity. We also used the T98G cell line derived from glioblastoma multiform tumor tissue. All glioma tumor cell lines were obtained from the American Type Culture Collection (ATCC, Manassas, VA, USA).

A172, H4, and U118-MG cell lines were cultured in the Dulbecco’s Modified Eagle’s medium supplemented with a 10% calf fetal serum and antibiotics (penicillin 100 units/mL and streptomycin 0.1 mg/mL) in 5% CO_2_ atmosphere at 37 °C. U87-MG and T98G cell lines were cultured in the Eagle’s minimum essential medium supplemented with a 10% calf fetal serum and antibiotics (penicillin 100 units/mL and streptomycin 0.1 mg/mL) in 5% CO_2_ atmosphere at 37 °C. SW1088 cell line was cultured in the Leibovitz’s L-15 medium supplemented with a 10% calf fetal serum and antibiotics (penicillin 100 units/mL and streptomycin 0.1 mg/mL) in 5% CO_2_ atmosphere at 37 °C. Cells were propagated every 3–4 days when they reached 70–80% of confluence using trypsin-EDTA solution (Sigma-Aldrich, St. Louis, MO, USA).

The human chronic myelogenous leukemia K562 cell line was cultured in the RPMI-1640 medium supplemented with a 10% calf fetal serum and antibiotics (penicillin 100 units/mL and streptomycin 0.1 mg/mL) in 5% CO_2_ atmosphere at 37 °C. Cells were obtained from a European collection of authenticated cell cultures (ECACC, Salisbury UK).

K562 cells with high and moderate expressions of ABCB1, K562/DoxDR3, and K562/DoxDR2 cells were derived from drug selected K562/Dox cells kindly provided by J.P. Marie (University of Paris 6, Paris, France). Detailed characterizations of K562/Dox, K562/DoxDR3, and K562/DoxDR2 cells are given elsewhere [30,44].

We further used K562/ABCG2CL10 and K562/ABCG2CL4 cells expressing high and moderate levels of ABCG2, respectively. These subclones were derived from the original K562/ABCG2 cell line by limiting dilution [31]. The K562/ABCG2 cell line was kindly provided by B. Sarkadi (National Blood Center and Semmelweis University, Budapest, Hungary) and the characterization of this cell line can be found elsewhere [31,45].

Resistant variants of K562 cells expressing ABCB1 or ABCG2 were cultured under the same conditions as maternal K562 cells (see above).

The cell density was assessed using an automatic analyzer Vi-CELL (Beckman Coulter, Indianapolis, IN, USA).

### 4.3. Cell Viability Assessment

Cell viability was determined by measuring cytoplasmic membrane integrity using the trypan blue exclusion assay on a Vi-CELL automated analyzer (Beckman Coulter, Indianapolis, IN, USA).

### 4.4. Cytotoxicity Assay

The reduction of 3-(4,5-dimethylthiazolyl)-2,5-diphenyl-tetrazolium bromide (MTT assay) was used as standard assay for the determination of in vitro cytotoxicity [46]. The MTT assay was performed after 120 h drug exposure on 96-well plates. The cytotoxic effects were expressed as IC_50_ (IC_20_) values from a minimum of five concentrations of the studied drug using the SigmaPlot 11.0 software package (Systat Software Inc., San Jose, CA, USA).

### 4.5. Western Blot Analysis of Drug Transporter Expression

Due to higher reliability, expression of ABC transporters was carried out using Western blot analysis [47]. Cell extracts were prepared as described previously [31]. Samples (equivalent of 30 μg protein) were analyzed by Western blot using monoclonal anti-P-gp (ABCB1) antibody produced in mouse, clone F4 (1:1000); affinity isolated anti-ABCG2 antibody produced in rabbit (1:2000), and monoclonal antiactin antibody produced in mouse, clone AC-40 (1:2000). All primary antibodies were purchased from Sigma-Aldrich (St. Louis, MO, USA). The signal was detected using horseradish peroxidase–conjugated secondary antibody (1:3000; Dako, Glostrup, Denmark). Products were visualized using enhanced chemiluminesence (ECL; Amersham, Little Chalfont, UK).

### 4.6. Functional Assay of ABCB1 and ABCG2 

Calcein acetoxymethyl ester (calcein AM; Molecular Probes, Eugene, OR, USA) accumulation and pheophorbide A (Sigma-Aldrich, St. Louis, MO, USA) accumulation were used as functional assays of ABCB1 and ABCG2, respectively [31]. Cells were loaded with appropriate probe and then analyzed by flow cytometry (Cytomics FC500; Beckman Coulter, Indianapolis, IN, USA), as described previously [31]. 

### 4.7. Analysis of Cell Cycle and Apoptotic Cells

Cell cycle progression and the identification of apoptotic cells (i.e., cells with hypodiploid DNA content) was carried out using flow cytometric analysis of DNA content according to [48,49].

Briefly, cells washed in PBS were stained in PBS containing propidium iodide (PI; 10 μg/mL), 0.1% Triton X-100, and RNase A (100 μg/mL) for 30 min and then analyzed on a Cytomics FC 500 System (Beckman Coulter) using MultiCycle software (P.S. Rabinovitch, University of Washington, Seattle, WA, USA). At least 10,000 cells in each sample were analyzed.

### 4.8. Morphological Analysis of Apoptosis

Cells fixed in 70% ethanol in PBS (*v*/*v*) were stained with 2-(4-Amidinophenyl)-6-indolecarbamidine dihydrochloride (DAPI; Sigma-Aldrich, St. Louis, MO, USA) and nuclear morphology was examined using an Olympus BX60 (Olympus, Hamburg, Germany) fluorescence microscope as described previously [50]. At least 300 cells were counted in each sample.

### 4.9. Measurement of Caspase3/7 Enzymatic Activity 

Caspase3/7 enzyme (DEVDase) activity was measured in cytoplasmatic extracts using fluorescent substrate Ac-DEVD-AMC. For details, see [51].

### 4.10. Cell Extracts Preparation 

Cells (at a density of 5 × 10^5^/mL) were incubated in the growth medium with GDN or 17-AAG for 3 h in 5% CO_2_ atmosphere at 37 °C. Subsequently, cells were separated from the growth medium by centrifugation through a layer of silicone oil as described previously [52,53]. Cell pellets were extracted using ice cold 4% (*v*/*v*) formic acid in 40% (*v*/*v*) methanol in water [54]. Cell extracts were clarified by centrifugation (40,000*g ×* 10 min at 4 °C) and diluted with 20% (*v*/*v*) methanol in water and analyzed by liquid chromatography coupled with a low-energy collision tandem mass spectrometer (LC/MS/MS).

### 4.11. LC/MS/MS Analysis of GDN

We used the method as described previously [54]. Briefly, separations were performed on a Polaris C18-A, 5 µm, 250 × 2.0 mm column (Varian Inc., Lake Forest, CA, USA) connected with a guard C18, 4.0 × 2.0 mm precolumn (Phenomenex, Torrance, CA, USA). The temperature of the column compartment was adjusted at 35 °C. Solvents used for separation were A (95% acetonitrile in 20 mM ammonium acetate, *v*/*v*) and B (20 mM ammonium acetate, pH 6.8). The flow rate was 300 μL/min with linear gradient elution from 0 to 3 min (20% to 90% of solvent A), from 3 to 4 min (90% of solvent A), from 4 to 5 min (90% to 20% of solvent A), and from 5 to 8 min (20% of solvent A). Sample injection volume was set at 10 µL. The effluent was introduced into the API 3200 triple quadrupole mass spectrometer (MDS SCIEX, Ontario, Canada) and electrospray ionization in negative ion mode was used for detection. The mass spectrometer was operated in the Precursor Ion Scan mode (Q1 quadrupole at range 558.5–560.0 amu, Q3 quadrupole at 516.1 amu, scan time 1 s). Ion spray probe parameters were optimized for GDN standard infusion: needle voltage −5500 V and source temperature 400 °C were adjusted. The declustering potential and the collision energy were set at −50 V and −30 eV, respectively. The instrument was operated in unit resolution. 

### 4.12. LC/MS/MS Analysis of 17-AAG

The UltiMate 3000 RS HPLC system (Dionex, Germering, Germany) was used. The analytical column HyperClone BDS C18, 150 × 2.0 mm (i.d.), 5 µm particle size (Phenomenex, Torrance, CA, USA) was applied for separations under optimized HPLC conditions: gradient elution of solvent A = 95% acetonitrile in 20mM ammonium acetate and solvent B = 20 mM ammonium acetate (pH 6.8) from 0 to 3 min (20% to 90% of solvent A), from 3 to 4 min (90% of solvent A), from 4 to 5 min (90% to 20% of solvent A), and from 5 to 8 min (20% of solvent A); a flow rate of 300 μL/min; injection volume of 10 µL; temperature of the column compartment was adjusted to 35 °C. The API 3200 triple quadrupole mass spectrometer (MDS SCIEX, Ontario, Canada) and electrospray ionization in positive ion mode was applied for detection. The mass spectrometer was operated in the Precursor Ion Scan mode (Q1 quadrupole at range 585.7–586.7 amu, Q3 quadrupole at 493.2 amu). Ion spray probe parameters were set for 17-AAG standard infusion: needle voltage +5500 V and source temperature 400 °C were optimized. The declustering potential and the collision energy were set at +21 V and +30 eV, respectively. The instrument was operated in unit resolution.

### 4.13. Statistical Analysis

Data are reported as the means ± S.D. Statistical analyses were performed using SigmaPlot 11.0 software package (Systat Software Inc., San Jose, CA, USA). Statistical significance was determined using the Student’s *t*-test for two group comparisons, and one-way ANOVA for more than two groups comparisons. We used * or § for a significant result (*p* < 0.05) and ** or §§ for the very significant result (*p* < 0.01).

## 5. Conclusions

In this study, GDN and 17-AAG exhibited cytotoxic effects in all studied glioma tumor cell lines irrespective of genetic alterations at sub-micro-molar and nanomolar concentrations. The cell death induced by both HSP90 inhibitors bore the morphological and biochemical hallmarks of apoptosis. GDN and 17-AAG are weak substrates of the main drug transporters ABCB1 and ABCG2. For this reason, we assume that the BBB would only partially compromise their cytotoxic effects.

## Figures and Tables

**Figure 1 pharmaceuticals-14-00107-f001:**
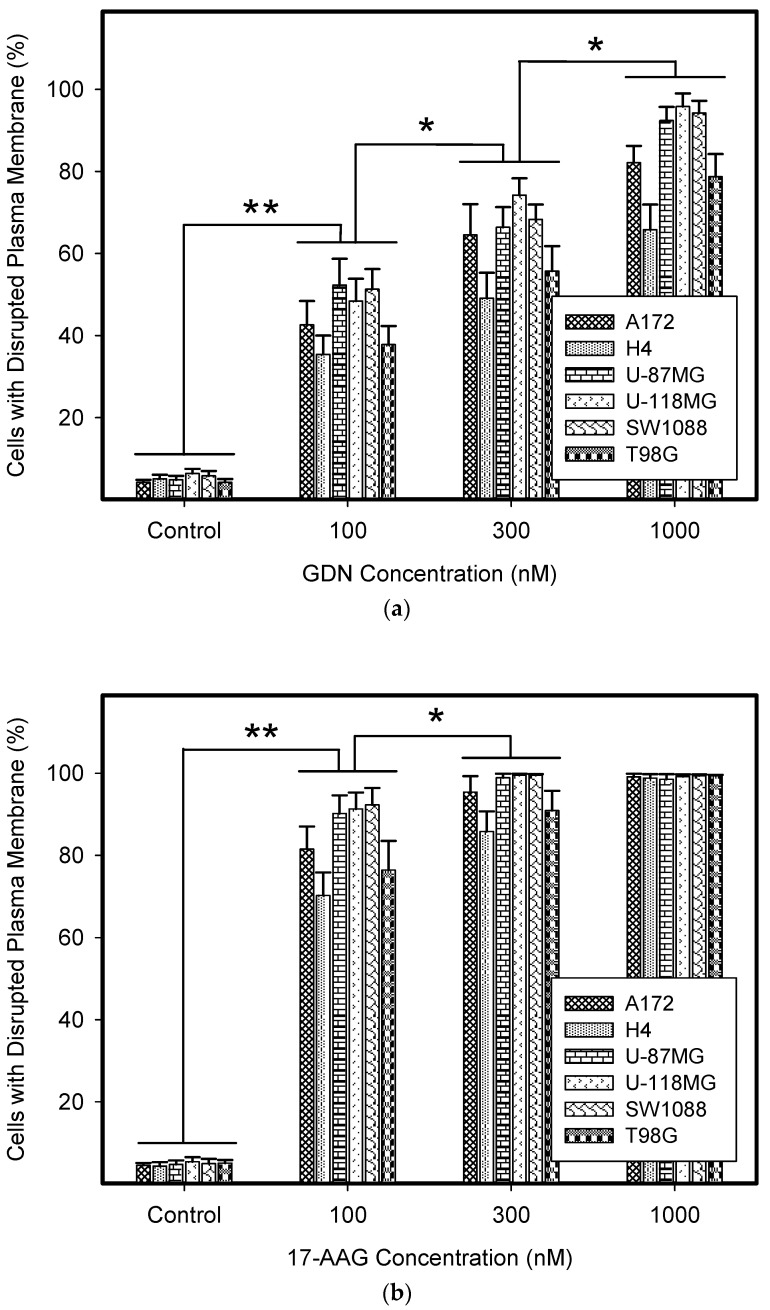
Effect of geldanamycin (GDN) and 17-allylamino-17-demethoxygeldanamycin (17-AAG) on cell death in glioma tumor cells. Cells were treated with GDN or 17-AAG for 5 days and then cell viability was assessed. (**a**) Effect of GDN on plasma membrane integrity in glioma tumor cells. (**b**) Effect of 17-AAG on plasma membrane integrity in glioma tumor cells. Columns represent the means of three independent experiments with S.D. * denotes significant change in the number of dead cells (*p* < 0.05) between the untreated (control) and treated groups of cells. ** denotes very significant change in the number of dead cells (*p* < 0.01) between the untreated (control) and treated groups of cells.

**Figure 2 pharmaceuticals-14-00107-f002:**
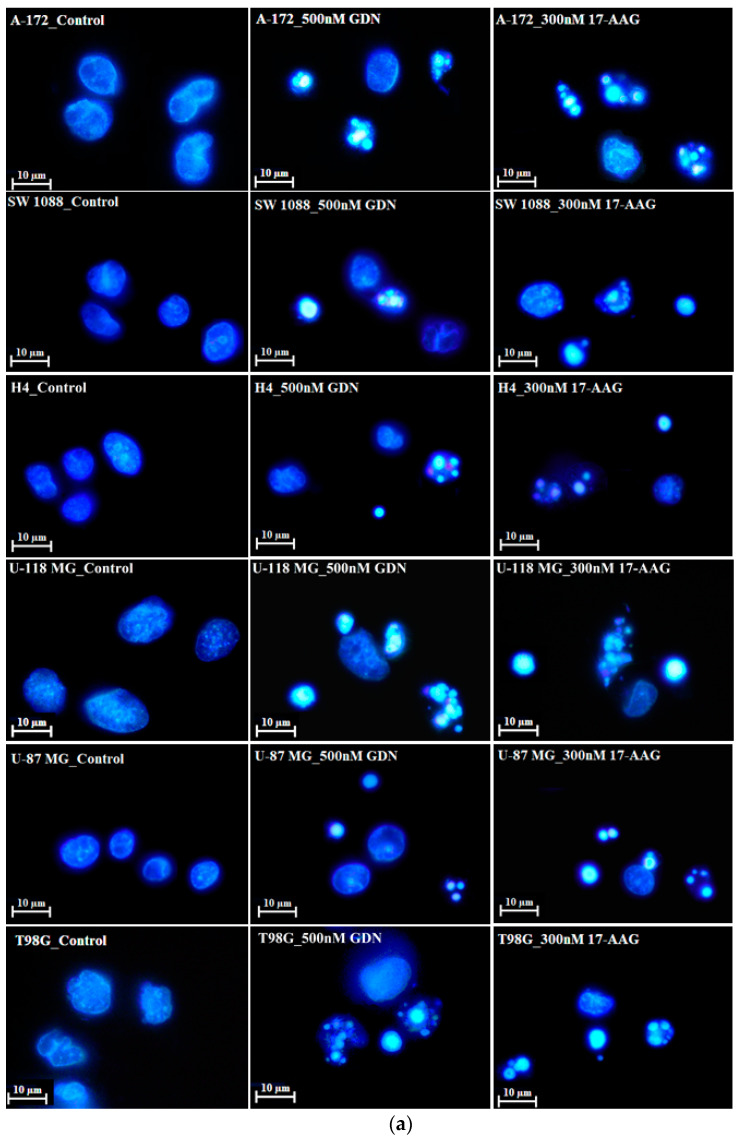
Effect of GDN and 17-AAG on apoptosis in glioma tumor cells. Cells were treated with GDN or 17-AAG for 3 days and then cells were subjected to appropriate assay. (**a**) Effect of GDN and 17-AAG on nuclear morphology in glioma tumor cells. Pictures represent typical examples. (**b**) Concentration-dependent effect of GDN on nuclear morphology in glioma tumor cells. (**c**) Concentration-dependent effect of 17-AAG on nuclear morphology in glioma tumor cells. Columns represent the means of three independent experiments with S.D. * denotes significant change in the number of cells with apoptotic nuclei (*p* < 0.05) between the untreated (control) and treated groups of cells. ** denotes very significant change in the number of cells with apoptotic nuclei (*p* < 0.01) between the untreated (control) and treated groups of cells. (**d**) Effect of GDN and 17-AAG on cell cycle progression and DNA fragmentation in glioma tumor cells. Histograms represent typical examples. (**e**) Effect of GDN and 17-AAG on capsase3/7 activation in glioma tumor cells. Columns represent the means of three independent experiments with S.D. * denotes significant change in caspase3/7 activity (*p* < 0.05) between the untreated (control) and treated groups of cells. ** denotes very significant change in caspase3/7 activity (*p* < 0.01) between the untreated (control) and treated groups of cells.

**Figure 3 pharmaceuticals-14-00107-f003:**
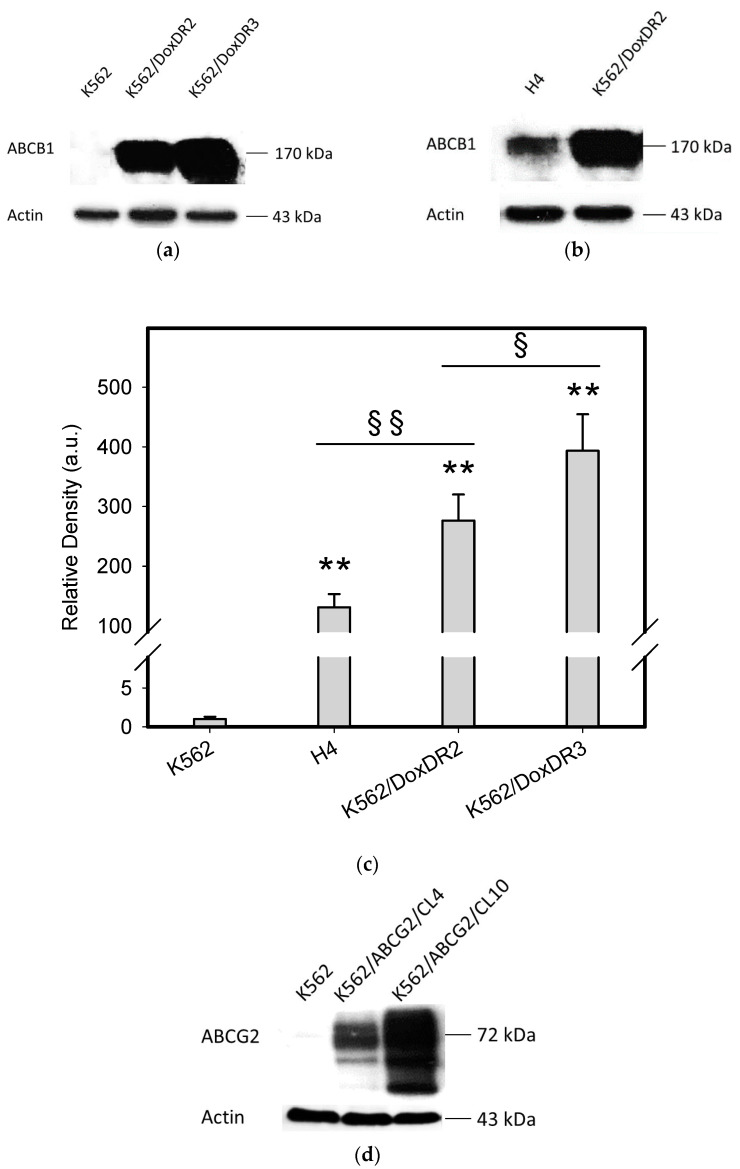
Analysis of main drug transporter expression. (**a**) Western blot analysis of ABCB1 expression in K562/DoxDR3 and K562/DoxDR2 cells. Picture represents typical result. (**b**) Western blot analysis of ABCB1 expression in H4 and K562/DoxDR2 cells. Picture represents typical result. (**c**) Quantitative analysis of ABCB1 expression. Columns represent mean from three independent experiments with S.D. ** denotes very significant change in ABCB1 expression (*p* < 0.01) between the sensitive K562 cells and resistant cells. § denotes significant change in ABCB1 expression (*p* < 0.05) between the resistant cells. §§ denotes very significant change in ABCB1 expression (*p* < 0.01) between the resistant cells. (**d**) Western blot analysis of ABCG2 expression in K562/ABCG2CL10 and K562/ABCG2CL4 cells. Picture represents typical result. (**e**) Quantitative analysis of ABCG2 expression. Columns represent mean from three independent experiments with S.D. ** denotes very significant change in ABCG2 expression (*p* < 0.01) between the sensitive K562 cells and resistant cells. §§ denotes very significant change in ABCG2 expression (*p* < 0.01) between the resistant cells.

**Figure 4 pharmaceuticals-14-00107-f004:**
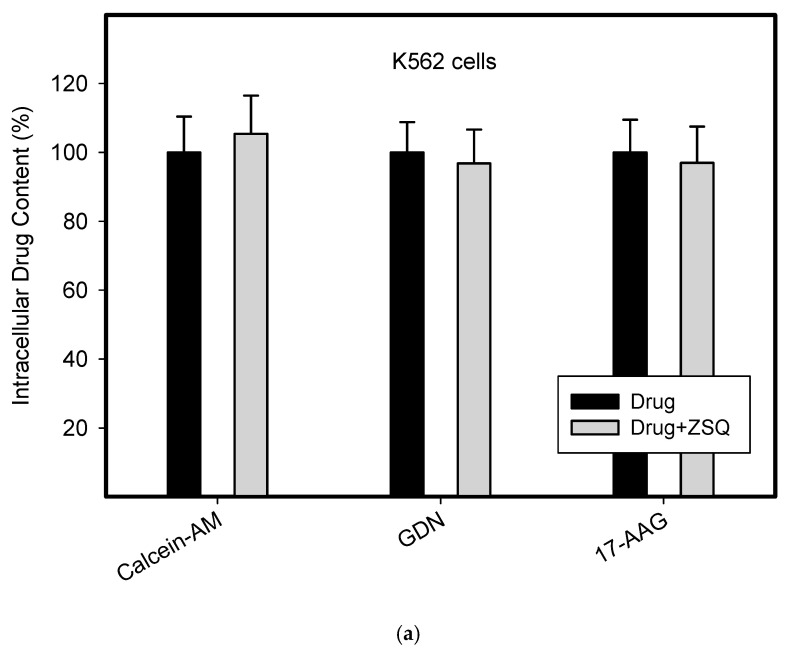
Effect of ABCB1 expression on intracellular drug levels in human leukemia cells. Cells were incubated for 3 h in the presence of 1µM GDN or 1µM 17-AAG with or without 0.5µM Zosuquidar (ZSQ). Cell extracts were analyzed using LC/MS/MS. Calcein-AM was used as reference substrate. Its content was determined using flow cytometry. (**a**) Intracellular drug levels in sensitive K562 cells. Intracellular levels of each drug in sensitive K562 cells was set to 100%. Columns represent mean from three independent experiments with S.D. (**b**) Intracellular drug levels in K562DoxDr2 cells with moderate ABCB1 expression. Columns represent mean from three independent experiments with S.D. ** denotes very significant change in intracellular drug level (*p* < 0.01) between the sensitive K562 cells and resistant K562DoxDr2 cells. §§ denotes very significant change between intracellular levels of indicated drugs (*p* < 0.01) in resistant K562DoxDr2 cells. (**c**) Intracellular drug levels in K562DoxDr3 cells with high ABCB1 expression. Columns represent mean from three independent experiments with S.D. ** denotes very significant change in intracellular drug level (*p* < 0.01) between the sensitive K562 cells and resistant K562DoxDr3 cells. §§ denotes very significant change between intracellular levels of indicated drugs (*p* < 0.01) in resistant K562DoxDr3 cells.

**Figure 5 pharmaceuticals-14-00107-f005:**
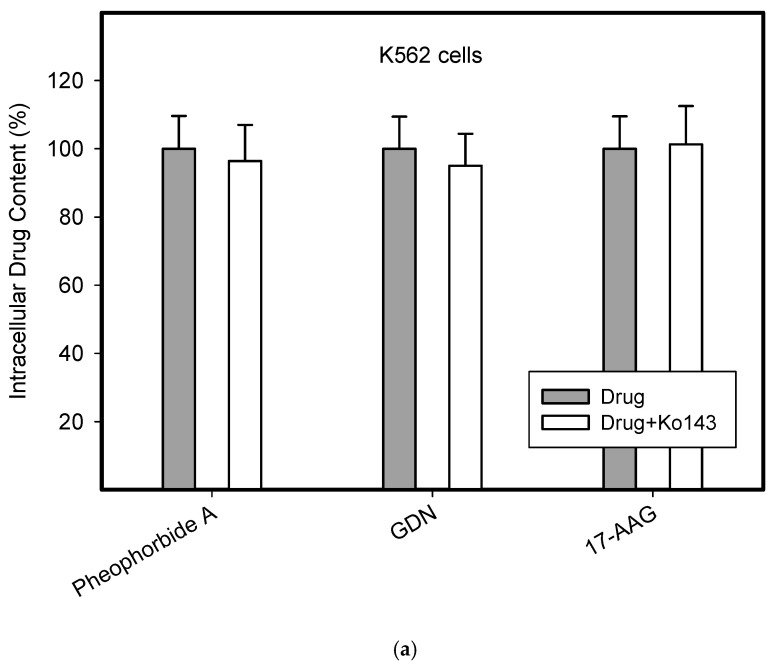
Effect of ABCG2 expression on intracellular levels of GDN and 17-AAG in human leukemia cells. Cells were incubated for 3 h in the presence of 1 µM GDN or 1 µM 17-AAG with or without 0.3 µM Ko143. Cell extracts were analyzed using LC/MS/MS. Pheophorbide A was used as reference substrate. Its content was determined using flow cytometry. (**a**) Intracellular drug levels in sensitive K562 cells. Intracellular levels of each drug in sensitive K562 cells was set to 100%. Columns represent mean from three independent experiments with S.D. (**b**) Intracellular drug levels in K562/ABCG2CL4 cells with moderate ABCG2 expression. Columns represent mean from three independent experiments with S.D. ** denotes very significant change in intracellular drug level (*p* < 0.01) between the sensitive K562 cells and resistant K562/ABCG2CL4 cells. §§ denotes very significant change between intracellular levels of indicated drugs (*p* < 0.01) in resistant K562/ABCG2CL4 cells. (**c**) Intracellular drug levels in K562/ABCG2CL10 cells with high ABCG2 expression. Columns represent mean from three independent experiments with S.D. * denotes significant change in intracellular drug level (*p* < 0.05) between the sensitive K562 cells and resistant K562/ABCG2CL10 cells. ** denotes very significant change in intracellular drug level (*p* < 0.01) between the sensitive K562 cells and resistant K562/ABCG2CL10 cells. §§ denotes very significant change between intracellular levels of indicated drugs (*p* < 0.01) in resistant K562/ABCG2CL10 cells.

**Table 1 pharmaceuticals-14-00107-t001:** Brief characterisation of glioblastoma cell lines used. Cells were obtained from American Type Culture Collection (ATCC).

Cell Line	ATCC^®^	Disease	Gene Mutation/Alteration
A172	CRL-1620	Glioblastoma	*CDKN2A*
			*CDKN2B*
			*EGFR vIII*
			*PTEN*
			*TP53*
H4	HTB-148	Neuroglioma	*CDKN2A*
			*CDKN2C*
			*MGMT down*
			*PTEN*
			*ABCB1 up*
U-87 MG	HTB-14	Likely Glioblastoma	*CDKN2A*
			*CDKN2B*
			*CDKN2C*
			*MGMT down*
			*PTEN*
U-118 MG	HTB-15	Glioblastoma Astrocytoma	*CDKN2A*
			*CDKN2B*
			*MGMT down*
			*PTEN*
			*TP53*
SW1088	HTB-12	Astrocytoma	*CDKN2A*
			*CDKN2B*
			*MGMT down*
			*PTEN*
			*TP53*
T98G	CRL-1690	Glioblastoma multiforme	*CDKN2A*
			*CDKN2C*
			*MGMT up*
			*PTEN*
			*TP53*

*ABCB1*—gene coding for P-glycoprotein; *CDKN2A*—cyclin-dependent kinase inhibitor 2A; *CDKN2B*—cyclin-dependent kinase inhibitor 2B; *CDKN2C*—cyclin-dependent kinase inhibitor 2C; *EGFR vIII*—epidermal growth factor receptor, variant III; *MGMT*—*O*-6-methylguanine-DNA-methyltransferase; *PTEN*—phosphatase and tensin homolog; *TP53*—tumor suppressor gene coding for p53 protein.

**Table 2 pharmaceuticals-14-00107-t002:** Cytotoxic effects of GDN and 17-AAG in glioblastoma cell lines.

Cell Line	GDN IC_50_ (nM)	GDN + ZSQ IC_50_ (nM)	17-AAG IC_50_ (nM)	17-AAG + ZSQ IC_50_ (nM)
A-172	214.7 ± 39.5	222.5 ± 42.8	34.8 ± 7.1	33.8 ± 6.6
H4	654.8 ± 147.5	183.9 ± 35.1	49.3 ± 8.0	28.2 ± 5.3
U-87 MG	104.4 ± 22.4	118.6 ± 27.8	21.4 ± 4.6	21.8 ± 4.3
U-118 MG	118.0 ± 20.2	121.5 ± 19.7	29.9 ± 5.2	28.2 ± 5.2
SW1088	124.5 ± 24.8	131.2 ± 20.4	35.6 ± 5.9	34.4 ± 7.2
T98G	549.6 ± 109.0	579.4 ± 91.2	40.4 ± 6.5	40.9 ± 7.9

Zosuquidar (ZSQ).

**Table 3 pharmaceuticals-14-00107-t003:** Cytotoxic effects of temozolomide (TMZ) and TMZ in combination with 10 µM lomeguatrib (LG) in glioblastoma cell lines.

Cell Line	TMZ IC_50_ (µM)	TMZ + LG IC_50_ (µM)	LG IC_20_ (µM)
A-172	54.4 ± 9.6	45.1 ± 9.9	>10
H4	78.8 ± 15.4	28.7 ± 6.1	>10
U-87 MG	26.4 ± 5.1	23.1 ± 4.4	>10
U-118 MG	71.3 ± 11.2	22.1 ± 3.7	>10
SW1088	51.1 ± 12.8	47.3 ± 8.0	>10
T98G	51.5 ± 9.9	49.1 ± 11.4	>10

## Data Availability

The data presented in this study are available on request from the corresponding author.

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
