# Peer review of "Heat Shock Protein Inhibitor 17-Allyamino-17-Demethoxygeldanamycin, a Potent Inductor of Apoptosis in Human Glioma Tumor Cell Lines, Is a Weak Substrate for ABCB1 and ABCG2 Transporters"

_pharmaceuticals, 2021, doi:10.3390/ph14020107_

Round 1

Reviewer 1 Report

No

Author Response

We thank the Reviewer for the second reading of our MS.

On the behalf of all authors,

Petr Mlejnek

Reviewer 2 Report

I appreciate the work done by the authors to improve their manuscript, however, the main problem with this research is a lack of novelty and use of the compounds which have several drawbacks.

In my opinion, the authors should be paid more attention to how they respond to reviewers. The reviewer’s duty is to show which aspects should be corrected to improve the research quality. Revision is also an opportunity for scientific discussion. The reviewers may not always correct the understanding of the authors’ intention. Thus, the authors may disagree with the reviewer’s suggestion/opinion; however, the authors’ response should be polite. The use of a statement such as “what a pity”, interjections, are quite rude.

The authors wrote that “the best of our knowledge we did not find a work demonstrating that GDN cannot cross the BBB. It is a pity that you do not substantiate your claim with the appropriate references.”

Please, could the authors read the following papers:

Lu, R. Ran, S. Parmentier‐Batteur, A. Nee, F.R. Sharp, Geldanamycin induces heat shock proteins in brain and protects against focal cerebral ischemia, J. Neurochem. 81 (2002) 355–364. https://doi.org/10.1046/j.1471-4159.2002.00835.x.

The authors mentioned that GA was injected into the lateral cerebral ventricles (i.c.v) because this large molecule was unlikely to pass the blood–brain barrier.

 J.R.C. Cha, K.J.H. St Louis, M.L. Tradewell, B.J. Gentil, S. Minotti, Z.M. Jaffer, R. Chen, A.E. Rubenstein, H.D. Durham, A novel small molecule HSP90 inhibitor, NXD30001, differentially induces heat shock proteins in nervous tissue in culture and in vivo, Cell Stress Chaperones. 19 (2014) 421–435. https://doi.org/10.1007/s12192-013-0467-2.

The authors stated that “because the pharmacokinetic/pharmacodynamic (PK/PD)profile and toxicity of geldanamycin make it unsuitable for in vivo administration, new compounds with reduced cytotoxicity and improved blood–brain barrier permeability are being developed”.

Regarding to 17-AAG the paper M.J. Egorin, E.G. Zuhowski, D.M. Rosen, D.L. Sentz, J.M. Covey, J.L. Eiseman, Plasma pharmacokinetics and tissue distribution of 17-(allylamino)-17-demethoxygeldanamycin (NSC 330507) in CD2F1 mice1, Cancer Chemother. Pharmacol. 47 (2001) 291–302. https://doi.org/10.1007/s002800000242.

In this paper, the authors stated that “analyses of tissue concentrations of17AAG  allowed the description  of  the  widespread  distribution  of the  drug,  the  relative  inability  of  17AAG  to cross the blood-brain barrier, and the relative exposures of tissues as opposed to plasma”

The authors in response to the reviewer, stated that "from this experimental finding, we conclude that these drugs could penetrate the BBB.", which findings the authors have in mind?

Author Response

Answer to Reviewer 2 (4)

I appreciate the work done by the authors to improve their manuscript, however, the main problem with this research is a lack of novelty and use of the compounds which have several drawbacks.

Response:

Thanks to the Reviewer for carefully analyzing our answers and other critical comments on our article. We have taken your comments into account and rewritten part of the Discussion according to your suggestions.

Changes are in green.

We are aware that our results do not represent “a revolution” in the treatment of GBM, however, we believe that we have contributed a bit to expanding knowledge in this area in two ways – i) studying the cell death mode across the glioma tumor cell panel and ii) quantitative relationships between ABCB1 transporter expression and intracellular levels of both HSP90 inhibitors. In addition, the observed interaction between ABCG2 and both HSP90 inhibitors is an original finding, to the best of our knowledge, no such work has been published.

The reasons why we used GDN and 17-AAG were given in the previous answers.

In my opinion, the authors should be paid more attention to how they respond to reviewers. The reviewer’s duty is to show which aspects should be corrected to improve the research quality. Revision is also an opportunity for scientific discussion. The reviewers may not always correct the understanding of the authors’ intention. Thus, the authors may disagree with the reviewer’s suggestion/opinion; however, the authors’ response should be polite. The use of a statement such as “what a pity”, interjections, are quite rude.

Response:

Thanks to the Reviewer for the patient discussion and we fully agree with your opinion! We definitely didn't want to offend you in any way. And if our wording of the answers has affected you in any way, we apologize! (I know from personal experience how ungrateful this work is!)

What we wanted to achieve was to talk about the specific arguments of specific researchers. Unfortunately, the scientific literature is sometimes full of conflicting conclusions / opinions, so it is difficult to orient oneself in them. And this certainly applies to HSP90 inhibitors and their possible application in the treatment of GBM.

The authors wrote that “the best of our knowledge we did not find a work demonstrating that GDN cannot cross the BBB. It is a pity that you do not substantiate your claim with the appropriate references.”

Please, could the authors read the following papers:

Lu, R. Ran, S. Parmentier‐Batteur, A. Nee, F.R. Sharp, Geldanamycin induces heat shock proteins in brain and protects against focal cerebral ischemia, J. Neurochem. 81 (2002) 355–364. https://doi.org/10.1046/j.1471-4159.2002.00835.x.

The authors mentioned that GA was injected into the lateral cerebral ventricles (i.c.v) because this large molecule was unlikely to pass the blood–brain barrier.

 J.R.C. Cha, K.J.H. St Louis, M.L. Tradewell, B.J. Gentil, S. Minotti, Z.M. Jaffer, R. Chen, A.E. Rubenstein, H.D. Durham, A novel small molecule HSP90 inhibitor, NXD30001, differentially induces heat shock proteins in nervous tissue in culture and in vivo, Cell Stress Chaperones. 19 (2014) 421–435. https://doi.org/10.1007/s12192-013-0467-2.

Response:

Thank you for listing specific publications. We did not read these and therefore we studied them carefully. They state exactly what you wrote. Unfortunately, these are only the statements of the mentioned authors, which are not substantiated by the results of the study or work. (Maybe it's an experience, but it's not published.)

We really haven't found work proving that GDN can't cross the BBB. However, we admit that such work may exist. On the contrary, we have found work that shows that GDN can overcome the BBB, albeit at higher concentrations:

Manaenko A, Fathali N, Chen H, Suzuki H, Williams S, Zhang JH, Tang J. Heat shock protein 70 upregulation by geldanamycin reduces brain injury in a mouse model of intracerebral hemorrhage.  Neurochem Int. 2010 Dec;57(7):844-50. doi: 10.1016/j.neuint.2010.09.001.

As we wrote above, this area represents conflicting opinions and difficult to find the truth ...

The authors stated that “because the pharmacokinetic/pharmacodynamic (PK/PD)profile and toxicity of geldanamycin make it unsuitable for in vivo administration, new compounds with reduced cytotoxicity and improved blood–brain barrier permeability are being developed”.

Regarding to 17-AAG the paper M.J. Egorin, E.G. Zuhowski, D.M. Rosen, D.L. Sentz, J.M. Covey, J.L. Eiseman, Plasma pharmacokinetics and tissue distribution of 17-(allylamino)-17-demethoxygeldanamycin (NSC 330507) in CD2F1 mice1, Cancer Chemother. Pharmacol. 47 (2001) 291–302. https://doi.org/10.1007/s002800000242.

In this paper, the authors stated that “analyses of tissue concentrations of17AAG  allowed the description  of  the  widespread  distribution  of the  drug,  the  relative  inability  of  17AAG  to cross the blood-brain barrier, and the relative exposures of tissues as opposed to plasma”

The authors in response to the reviewer, stated that "from this experimental finding, we conclude that these drugs could penetrate the BBB.", which findings the authors have in mind?

Response:

The question of whether 17-AAG can cross the BBB is again somewhat controversial.

Nevertheless, the results of some authors indicate that 17-AAG can cross the BBB [1,2].

[1] Sauvageot, C.M.; Weatherbee, J.L.; Kesari, S.; Winters, S.E.; Barnes, J.; Dellagatta, J.; Ramakrishna, N.R.; Stiles, C.D.; Kung, A.L.; Kieran, M.W.; Wen, P.Y. Efficacy of the HSP90 inhibitor 17-AAG in human glioma cell lines and tumorigenic glioma stem cells. Neuro. Oncol. 2009, 11, 109-121.

Citation [1] was mentioned in previous version of our MS.

[2] Waza M, Adachi H, Katsuno M, Minamiyama M, Sang C, Tanaka F, Inukai A, Doyu M, Sobue G. 17-AAG, an Hsp90 inhibitor, ameliorates polyglutamine-mediated motor neuron degeneration.  Nat Med. 2005 Oct;11(10):1088-95. doi: 10.1038/nm1298.

The citation [2] was added to the Discussion and References. Please, se revised version of our MS.

As far as the work of Egorin et al is concerned, you are right that the authors wrote about the rapid distribution of 17-AAG to all tissues except the brain. Therefore, we rewritten this part of Discussion. Please, see revised version of our MS.

However, the data presented in the work above show that the distribution in brain tissue is not insignificant, but is approximately 3-5 times lower than in plasma and 10-100 times lower than in other tissues (Table 5 and 8). But still this value is in order of micromoles and is approximately 20 times greater than our IC50 values determined ​​for all studied glioma tumor cells. Although the concentration of 17-AAG in brain tissue is relatively low, the possibility of a cytotoxic effect on tumor cells in the brain cannot be clearly ruled out. And as we also wrote in the section Discussion, it is necessary to take into account the substrate specificity of ABC transporters, which may not be the same in humans and mice.

Reviewer 3 Report

No additional comments

Author Response

We thank the Reviewer for the second reading of our MS.

On the behalf of all authors,

Petr Mlejnek

This manuscript is a resubmission of an earlier submission. The following is a list of the peer review reports and author responses from that submission.

Round 1

Reviewer 1 Report

The manuscript "Heat shock protein inhibitor, 17-allyamino-17-demethoxygeldanamycin, that is a very potent inductor of apoptosis in human glioma tumor cell lines, is a weak substrate for ABCB1 and ABCG2 transporters" from Pastvova et al., describes the cytotoxic effects of 17AAG and GDN on different GBM cell lines and their potential efflux via specific transporters. Given the high mortality associated with GBM the study of anti-tumor agents is of interest for cancer biology and therapy.

Minor comments:

1. The study is sound and of good quality, but there is a lack of novelty that partially affect the overall manuscript. In general, the use of HSP90 inibitors to treat cancers has been very promising in vitro for many types of cancers, but with poor results in the clinic. So far, at least 18 different inhibitors have been evaluated in clinical trials, but none have shown satisfactory efficacy to be approved by the FDA. The main reason for the limited efficacy can be attributed to the dose-limiting toxicity at the drug dose of near-complete client depletion. In addition, the activation of HSF1 lead to cytoprotective heat shock response.

I suggest the Authors to better discuss this limit in relation with GBM.

For example, novel developments of HSP90 inhibitors (Park et al. Experimental & Molecular Medicine (2020) 52:79–91) explain a potential novel pan-HSP90 inhibitor that may overcome some of the limitations of 17-AAG.

2. please add molecular weight on the WB in Figure 3

Author Response

Please see attached response letter.

Reviewer 2 Report

The authors found that GDN and 17-AAG exhibited strong proapoptotic effect in glioma cells. Moreover, GDN and 17-22 AAG appeared to be weak substrates of ABCB1 and ABCG2. However, several reports showed GDN or 17-AAG induced apoptosis in glioma cells (Cancer Res. 2001 May 15;61(10):4010-6.; J Cell Physiol. 2004 Dec;201(3):374-84. ;Neuro Oncol. 2009 Apr;11(2):109-21.). GDN is a substrates of ABCB1(Pharm Res. 2007 Sep;24(9):1702-12.). Therefore, the novelty of this study is low. Furhermore, the apoptotic nuclei of Fig 2a and sub-G1 of Fig 2d are unconsistent. The concentrations of ZSQ in Fig 4 and Ko143 in Fig 5 did not shown.

Author Response

Please see attached response letter.

Reviewer 3 Report

In this manuscript, Pastvova et al. evaluate the effects of GDN and 17-AAG in proapoptotic action in glioma cell lines irrespective of genetic alterations. Authores suggested that, GDN and 17-AAG appeared to be weak substrates of ABCB1 and ABCG2. Moreover, study team hypothesize that GBM patients may benefit from 17-AAG either as a single agent or in combination with other drugs.The manuscript is easy to read, follow and understand. Methods are clear and provide enough data to reproduce the experiments. Main limitation is an "in vitro" type of study - which should me mentioned in limitation section. All conclusion should be confirmed by  "in vivo" studies in the future.

Introduction:

lines 43-46 Cite:

Litak J, Grochowski C, Litak J, Osuchowska I, Gosik K, Radzikowska E, Kamieniak P, Rolinski J. TLR-4 Signaling vs. Immune Checkpoints, miRNAs Molecules, Cancer Stem Cells, and Wingless-Signaling Interplay in Glioblastoma Multiforme-Future Perspectives. Int J Mol Sci. 2020 Apr 28;21(9):3114. doi: 10.3390/ijms21093114. PMID: 32354122; PMCID: PMC7247696.

Discussion:

Add study limitation at the end of discussion. Cell lines study as a main thing.

In my opinion the classic layout of the article with the materials and methods before the discussion, seems more appropriate.

Spell-check is required.

Change GMB for GBM lines 24, 51,279

All abbreviations should be explained when they first appear in the text.

Author Response

Please see attached response letter.

Reviewer 4 Report

The presented paper described the anticancer activity of geldanamycin and its analog 17-AAG towards glioblastoma cell lines. The authors mainly focused on the role of ABCB1 and ABCG2 in the resistance mechanisms. This paper possesses some major flaws, which are described below.

- The manuscript needs to be carefully reviewed because several sentences are incorrect and/or contain typographical or structural errors.

- The tile is too long and should be changed.

- The main problem with this research is that GDN and 17-AAG are not a “weak” substrate of ABCB1. The conclusion in lines 263-272 about ABCB1 is not accurate. Based on the data, cells expressed ABCB1 are significantly resistant to GDN and 17-AAG because the decrease in the cellular level of 45-65% may affect tested compounds’ activity. Moreover, data for the H4 cell line rather confirm these concerns. The authors found that the GDN combined with ZSQ decrease IC50 around 3.5 times (IC50 of 183.9 nM compare to over 600 nM for GDN alone). Thus, it seems that GDN as a substrate of ABCB1 may exert lower activity. Could the authors look carefully again at these data and discussed it again?

- It was previously found that GDN and 17-AAG are substrates for ABCB1; moreover, some studies showed that 17-AAG may inhibit ABCB1; thus could the authors explain the novelty of their research?

- The authors hypothesized that “17-AAG can cross the BBB at concentrations that may be cytotoxic to GMB”. Could the authors explain the reason for this hypothesis?

- The authors wrote that “We searched for a drug whose molecular target is common to all tumors and which can cross BBB, or at least its interactions with main drug transporters must be weak.” However, it is well-known that geldanamycin cannot cross the blood-brain barrier.

- Could the authors discuss how these results may be transferred to clinical situations?

- What is very important, and I could not find in this article is the concentration range of temozolomide, LG, and ZSQ used to obtain IC50.

- In general, all the figures should be changed. It is very hard to follow the findings. I suggest changing orientation to landscape.

- In general, the results are very brief, and the findings are often not clear. Please review the data and highlight any findings that could be potentially interesting, supplemented with a clear rationale on why these findings are interesting.

- Figure 1: the authors should plot their data as dose-response curves with appropriate fits. Did the authors use three concentrations to calculate IC50? How the authors calculate the IC50? Why the authors used trypan blue staining to determine IC50? It will be more accurate to use more sensitive assays such as luminescence-based assay.

- Figure 2: there are several mistakes in the figure’s caption. The authors described panels a,b, and c, while the figure contains panels from A to E. - These histograms should reveal cells’ distribution in three major phases of the cycle (G0/G1, S, and G2/M).

- How did the authors select the dose of TMZ for their studies?

- The abbreviation of Zosuquidar trihydrochloride (ZSQ) should be explained in a place where it appears for the first time.

- The explanation of the reason to use ZSQ should be added. Zosuqidar is a potent inhibitor of P-glycoprotein; however, there is no information about this.

- The same concern is for lomeguatrib and Ko-143. Some readers may not know that LG is an inactivator of O(6)-methylguanine-DNA methyltransferase, and Ko-143 is an inhibitor for ABCG2. This information should be added.

- The authors write that “Given that GDN cannot be used in clinics due to its side effects”; thus could the authors explain why they tested GDN? Maybe it will be better to use a novel Hsp90 inhibitor?

- Lines 72-75: this statement needs at least one reference.

Author Response

Please see attached response letter.
